# Enhanced Propanol Response Behavior of ZnFe$_2$O$_4$ NP-Based Active Sensing Layer Induced by Film Thickness Optimization

**Murendeni I. Nemufulwi** [1,2], **Hendrik C. Swart** [2] and **Gugu H. Mhlongo** [1,2,*]

1  Centre for Nanostructures and Advanced Materials (CeNAM), DSI-CSIR Nanotechnology Innovation Centre, Council for Scientific and Industrial Research, Pretoria ZA0001, South Africa; mnemufulwi@csir.co.za
2  Department of Physics, University of the Free State, Bloemfontein ZA9300, South Africa; swarthc@ufs.ac.za
*  Correspondence: ghmhlongo@csir.co.za; Tel.: +27-12-841-3935

**Abstract:** Development of gas sensors displaying improved sensing characteristics including sensitivity, selectivity, and stability is now possible owing to tunable surface chemistry of the sensitive layers as well as favorable transport properties. Herein, zinc ferrite (ZnFe$_2$O$_4$) nanoparticles (NPs) were produced using a microwave-assisted hydrothermal method. ZnFe$_2$O$_4$ NP sensing layer films with different thicknesses deposited on interdigitated alumina substrates were fabricated at volumes of 1.0, 1.5, 2.0, and 2.5 µL using a simple and inexpensive drop-casting technique. Successful deposition of ZnFe$_2$O$_4$ NP-based active sensing layer films onto alumina substrates was confirmed by X-ray diffraction and atomic force microscope analysis. Top view and cross-section observations from the scanning electron microscope revealed inter-agglomerate pores within the sensing layers. The ZnFe$_2$O$_4$ NP sensing layer produced at a volume of 2 µL exhibited a high response of 33 towards 40 ppm of propanol, as well as rapid response and recovery times of 11 and 59 s, respectively, at an operating temperature of 120 °C. Furthermore, all sensors demonstrated a good response towards propanol and the highest response against ethanol, methanol, carbon dioxide, carbon monoxide, and methane. The results indicate that the developed fabrication strategy is an inexpensive way to enhance sensing response without sacrificing other sensing characteristics. The produced ZnFe$_2$O$_4$ NP-based active sensing layers can be used for the detection of volatile organic compounds in alcoholic beverages for quality check in the food sector.

**Keywords:** sensing layer; zinc ferrites; gas sensing; nanoparticles; drop casting

## 1. Introduction

Volatile organic compounds (VOCs) emitted from food products are essential as they give rise to different aromas and serves as indicators for food freshness [1,2]. Aromas are mainly crucial for a distinctive taste and making food more enjoyable. Additionally, detection of VOCs making up such aromas is significant for classification purposes. Research studies have shown that a range of VOCs has been widely detected in wines and alcoholic beverages for quality checks [3,4]. In particular, wine aroma and its analysis is a significant parameter responsible for quality and consumer acceptance. Conventional methods such as gas chromatography have been widely used for the analysis and classification of wines. Lukic et al. [3] previously used gas chromatographic analysis of minor aromas in wines for sensory evaluation. Their study showed that it is possible to use gas chromatography for wine classification. However, sensory evaluation by chemical analysis requires a large set of samples for accurate and reliable classification and prediction models. In general, conventional methods require laboratory settings and skilled personnel [5]. The high cost associated with gas chromatography and extensive laboratory steps limits them from real-time and onsite applications. On the other hand, semiconducting metal oxides (SMOs) have attracted much attention owing to their simple fabrication, portability, and low cost [6].

Moreover, SMOs offer high sensitivity, selectivity, simple operation, and stable properties, making them ideal in various applications. In addition, the easy integration of SMOs in different devices allows them to be incorporated with different technologies. Over the past decades, several SMOs such as $WO_3$ [7], ZnO [8], $LaCoO_3$ [9], $TiO_2$ [10], and $ZnFe_2O_4$ [11–13] have been widely researched and developed to detect various VOCs. Despite the variety of available SMOs, researchers continue to search for sensitive, selective, and stable sensing materials with low operating temperatures. Among them, $ZnFe_2O_4$ readily detects VOCs, making it a promising SMO for classification and quality check sensors. However, the sensitivity, selectivity, and high operating temperature of $ZnFe_2O_4$ need to improve for its use in practical applications.

Researchers have focused on improving the sensing properties of $ZnFe_2O_4$ through its electrochemical properties to address these setbacks. This is conducted by the development of novel synthesis procedures [14], surface modification [5,15], cation substitution, [16,17], and heterostructure formation [18,19]. In contrast, developing sensitive layer deposition methods to attain ideal sensors suitable for prototype systems is significant. Previous findings have also demonstrated that gas sensing takes place at different sites of the structure depending on morphology [20]. Thus, the orientation of morphology on the deposited sensitive layer highly influences the gas exposure to different sites. Therefore, different chemical and physical deposition techniques to produce sensitive films of $ZnFe_2O_4$ have been adopted. However, such deposition techniques normally produce compact layers (thin films) with gas interaction only at the geometric surface. Further, these techniques are complicated to use and sometimes require high processing power [21]. On the other hand, a simple drop-casting method can produce a porous layer where the volume of the layer is also accessible to the gas, creating a much higher active surface than the geometric one [22]. In addition, mass transportation of the gas and diffusion over the entire sensing layer can lead to improved sensing characteristics [23].

Although the drop-casting method has been used before to prepare active $ZnFe_2O_4$ sensitive layers, the depositing parameters have often been neglected leading to poor sensing characteristics and reproducibility issues [24,25]. The dispersion of the nanostructures deposited on a substrate to produce an active sensing layer is influenced by the volumetric amount and evaporation rate of the solution which in turn affects the sensing characteristics of the sensitive layer. Thus, for reproducible sensitive surfaces, the radial outward flow of the drops forming the sensitive layer needs to be controlled by the volume deposited and evaporation rate of the solution [26]. While optimization of sensing film's thickness can lead to enhanced sensing characteristics, this often comes with an expense of sacrificing crucial sensing characteristics. Barreca et al. [27] realized this when their chemical vapor-deposited $MoO_3$-$Bi_2O_3$ thin films possessed a long response time of 5 min towards methane with no stability and repeatability characteristics. Korotcenkov et al. [28] further explained the negative influence of highly porous layers to time constants of $In_2O_3$ sensing films. Based on this, development of reliable sensing fabrication approach is still needed.

Therefore, this work focuses on the optimization of the $ZnFe_2O_4$ NP-based sensing film thickness through an inexpensive drop-casting method. Sensing films with different thicknesses deposited on interdigitated alumina substrates were fabricated at volumes of 1.0, 1.5, 2.0, and 2.5 µL. The suitability to use a $ZnFe_2O_4$ NP-based sensor with an optimized film thickness for the detection of VOCs in alcoholic beverages for quality check in the food sector has been evaluated through comparison of gas sensing performance studies at an operating temperature of 120 °C. The $ZnFe_2O_4$ NP-based sensor fabricated at a volume of 2 µL displayed better sensing characteristics while maintaining good response and recovery times. A gas sensing mechanism describing the observed enhanced gas sensing behavior has been proposed.

## 2. Materials and Methods

Zinc (II) nitrate hexahydrate ($Zn(NO_3)_2 \bullet 6H_2O$), iron (III) nitrate nonahydrate ($Fe(NO_3)_3 \bullet 9H_2O$), sodium hydroxide (NaOH), α-Terpineol ($C_{10}H_{18}O$), and cellulose were

all purchased from Sigma-Aldrich (South Africa). All chemicals were used as received without any further purification. Distilled water used in the aqueous solutions was prepared with Mili-Q water.

A microwave hydrothermal synthesis method was used for the preparation of the $ZnFe_2O_4$ NPs. In a typical procedure, a 1:2 mole ratio of $Zn(NO_3)_2 \bullet 6H_2O$ and $Fe(NO_3)_3 \bullet 9H_2O$ was dissolved in 50 mL of distilled water. The precursor solution was stirred for an hour using a magnetic stirrer at room temperature. NaOH base precursor (0.5 M) was added dropwise to the solution until a pH of 7 was reached. The solution was further stirred for 10 min before being transferred to Teflon vessels placed in a digestive microwave. The solution was irradiated at 150 °C for 15 min and then allowed to cool down. The produced precipitate was washed six times using distilled wasted and absolute ethanol, and a centrifuge collected the precipitate. The precipitate was then dried in an oven at 80 °C for 12 h and annealed at 500 °C for 3 h to archive crystallinity. Finally, a pestle and mortar were used to obtain nanostructured $ZnFe_2O_4$ in the powder form.

The illustration of the deposition of $ZnFe_2O_4$ NP-based sensing films is presented in Figure 1. To produce homogeneously dispersed $ZnFe_2O_4$ NPs, the cellulose was dissolved in terpineol at a mole ratio of 1:10, respectively. This solution was kept at 60 °C on a magnetic stirrer until the cellulose was dissolved completely. 0.03 g of $ZnFe_2O_4$ product was added in 0.25 mL of the terpineol/cellulose solution, which was then ultrasonicated for 1 h. A micropipette was used to drop cast the sonicated mixture onto commercially purchased interdigitated alumina $(Al_2O_3)$ substrates at volumes of 1.0, 1.5, 2.0, and 2.5 µL. The produced sensing films were dried in an oven for 1 h and further annealed at 300 °C for 2 h. The gas sensing capabilities of the fabricated $ZnFe_2O_4$ based gas sensors were evaluated using a KSGAS6S gas sensing station (KENOSISTEC, Brescia, Italy) at operating temperatures ranging from 25 to 180 °C under dry atmospheric conditions with a source voltage of 2.0 V applied across the sensors. The operating temperature was determined by varying the applied voltage across the sensor's heater with a thermocouple used to measure the resultant temperature. The desired concentration of target gases (ppm) of methanol, propanol, ethanol, methane, and carbon monoxide were varied at flow ratios with synthetic air in a sensing chamber. The sensors output was in current (mA) which was converted to resistance for response calculations. The response was calculated by $R = R_a/R_g$ for all the target reducing gases, where $R_a$ is the resistance in air and $R_g$ the resistance in the reducing gas. X-ray diffraction (PANalytical Xpert PRO) using a CuKα radiation source (λ = 0.15406 nm) was used for phase identification and structural analysis. Surface morphology and cross-sectional analysis were conducted using a field-emission scanning electron microscope (FE-SEM, Auriga, and Zeiss, Germany). Elemental mapping of the sensing layers was carried out using an energy dispersive spectrometer (EDS) coupled with an FE-SEM.

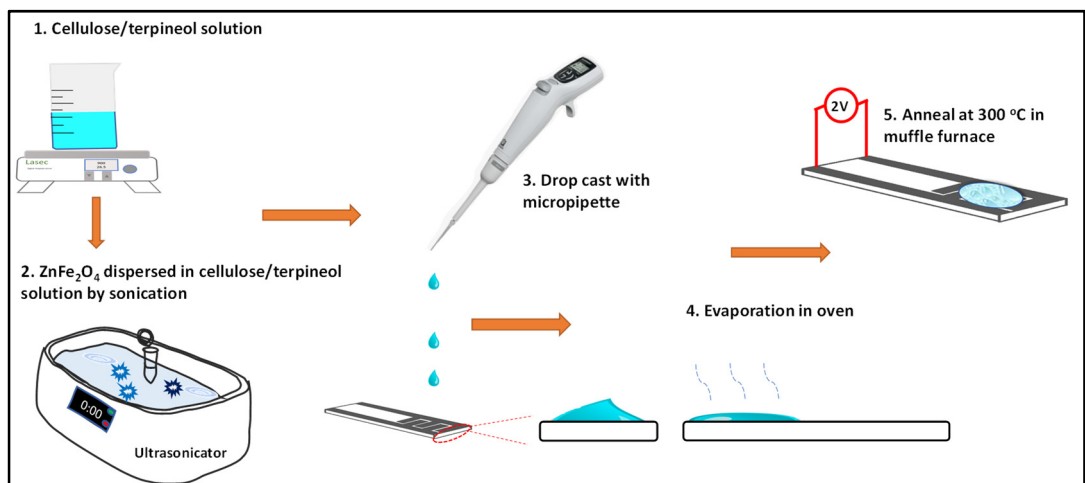

**Figure 1.** Fabrication process followed to produce $ZnFe_2O_4$ NP-based sensing layer films using a drop-casting method.

## 3. Results

### 3.1. Characterization

Figure 2 presents the XRD patterns of $ZnFe_2O_4$ NP-based sensing films deposited into interdigitated $Al_2O_3$ substrates. The $ZnFe_2O_4$ deposited sensing films at volumes of 0.5, 1.0, 1.5, 2.0, and 2.5 µL represented by bullet-labeled diffraction peaks can be indexed to the face-centred cubic structure (JCPDS file (card No. 86-0507). This indicates that the $ZnFe_2O_4$ NPs crystallized in a face-centered cube with no other impurity phases formed. The remaining diffraction peaks were perfectly indexed to the JCPDS file (card No. 83-2081) of $Al_2O_3$. It can be seen from the XRD patterns of all the $ZnFe_2O_4$ deposited sensing films that the diffraction peak intensity of $ZnFe_2O_4$ is weaker as compared to that of the $Al_2O_3$ substrate, indicating that the deposited $ZnFe_2O_4$ NP-based sensing film is significantly thinner compared to the substrate.

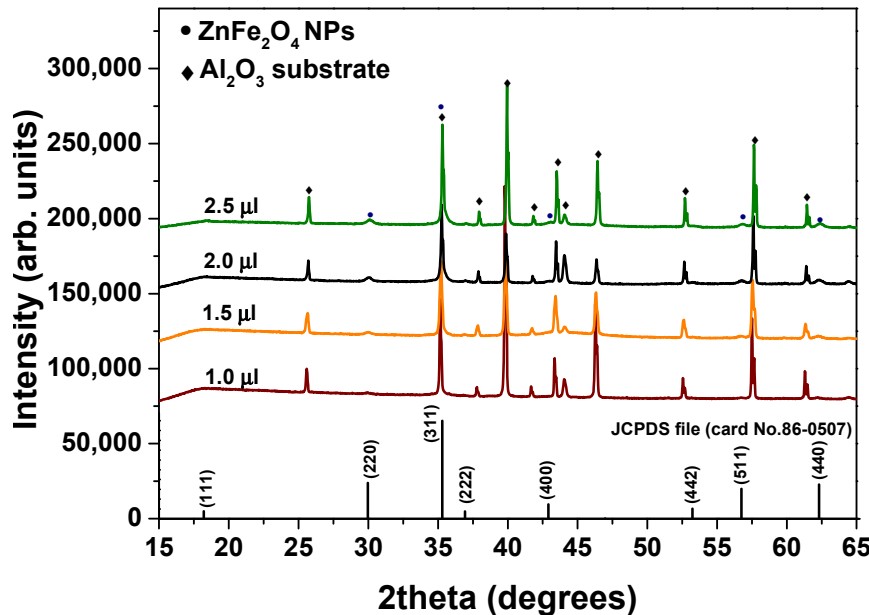

**Figure 2.** Diffraction patterns of the $ZnFe_2O_4$ deposited sensing films at volumes of 1.0, 1.5, 2.0, and 2.5 µL.

Figure 3a–e illustrates the cross-sectional and top-view SEM images of the $ZnFe_2O_4$ NP deposited sensing films at volumes of 1.0, 1.5, 2.0, and 2.5 µL, respectively. As displayed in this figure, the sensing film thickness values were found to be 3.92, 4.38, 5.15, and 5.61 µm for 1.0, 1.5, 2.0, and 2.5 µL volumes, respectively. The thickness varied with the deposited volumes and could be determined by the radial outward flow and evaporation rate of the deposited solution [26]. The cross-sectional images also display evidence of inter-agglomerate pores within the sensing layers. A higher magnification top view of the sensing films confirms the inter-agglomeration pores, which are prominent on the $ZnFe_2O_4$ sensing films prepared at volumes of 1.5 and 2.0 µL (Figure 3f,g, respectively). The top view images further indicate that the $ZnFe_2O_4$ sensing films were made of nanoparticles. The sensing film deposited at 1.0 µL in Figure 3e shows that the particles were very clustered with a smooth surface. However, as the sensing film volume was increased to 1.5 and 2.0 µL, the particles became dispersed with a rough surface showing inter-grain pores. The sensing film deposited at a volume of 2.5 µL started showing compact particles on the surface and inter-agglomerate pores were no longer visible.

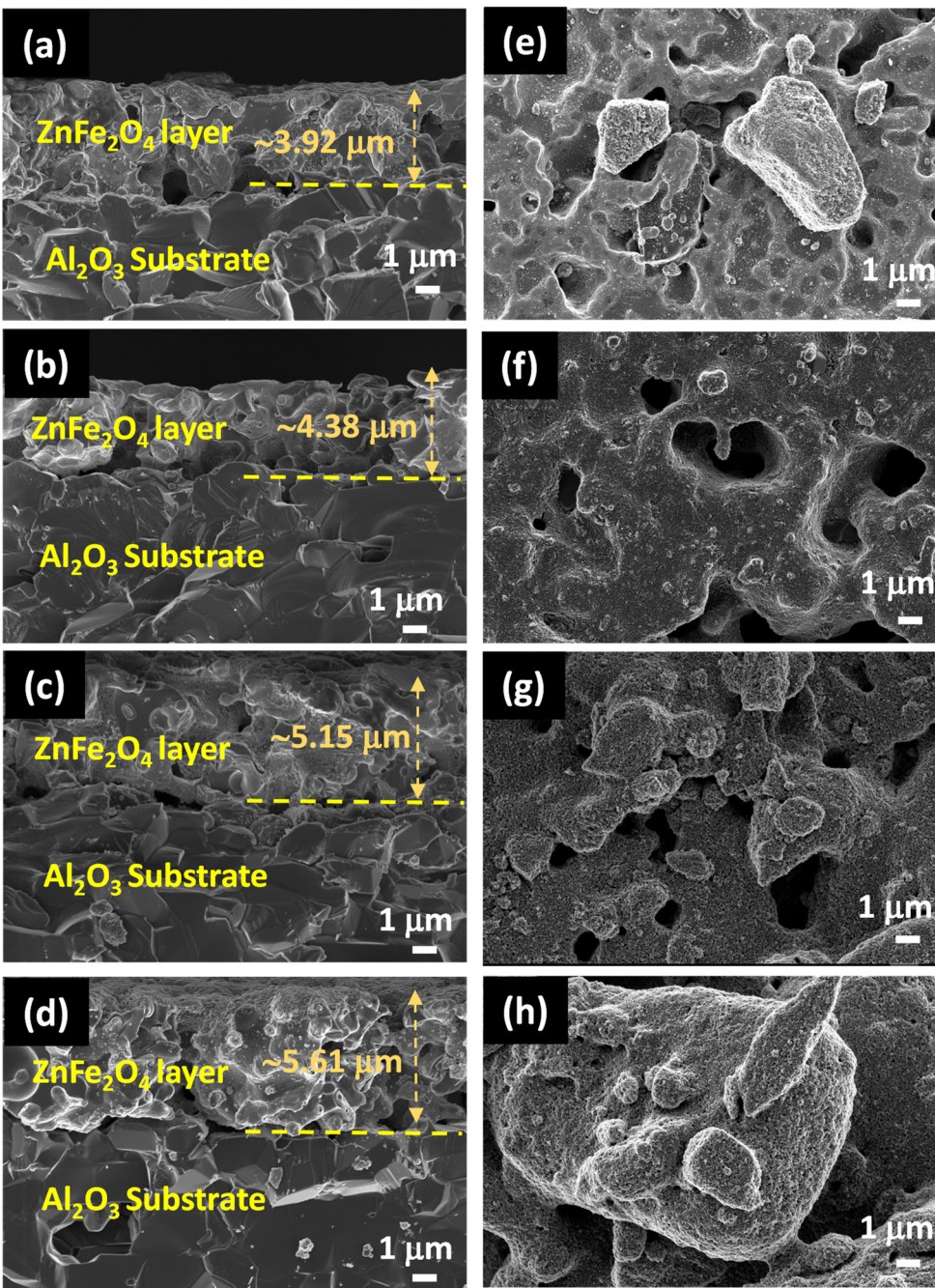

**Figure 3.** (**a**–**d**) Cross-sectional images of the $ZnFe_2O_4$ deposited sensing films at volumes of 1.0, 1.5, 2.0, and 2.5 μL, respectively, and (**e**–**h**) the corresponding top view SEM images.

Figure 4 presents the cross-sectional image view of $ZnFe_2O_4$ sensing film prepared at volume of 2.0 μL. A high-magnification top view SEM image presented in Figure 4b shows a uniform and well-dispersed distribution of $ZnFe_2O_4$ NPs. Figure 4c shows the $ZnFe_2O_4$ microscopic agglomerates distributed across the interdigitated electrodes. The microscopic distribution of the agglomerates was also visible to the naked eye, suggesting that the radial outward flow of the dropped solution was sufficient to cover the interdigitated electrodes. This is a consequence of the coffee ring effect, where particles form ring-like patterns on the surface/substrate [26]. As displayed in Figure 4d, the elemental maps show abundance of different elements ubiquitously distributed over the $Al_2O_3$ interdigitated electrodes. The compositional variation on the surface was further analyzed by EDS presented in Figure 5a–d for the sensing film prepared at volumes of 1.0, 1.5, 2.0, and

2.5 μL, respectively. An intensity of Al observed between the electrodes decreased with an increase of deposited volumes confirming different sensing film thickness. Intensities of Zn, Fe, and O displayed an increase with an increase of the deposited volumes. A stronger intensity increase of O is observed in between the electrodes, and this is contributed by O from $Al_2O_3$ substrate and $ZnFe_2O_4$ NPs.

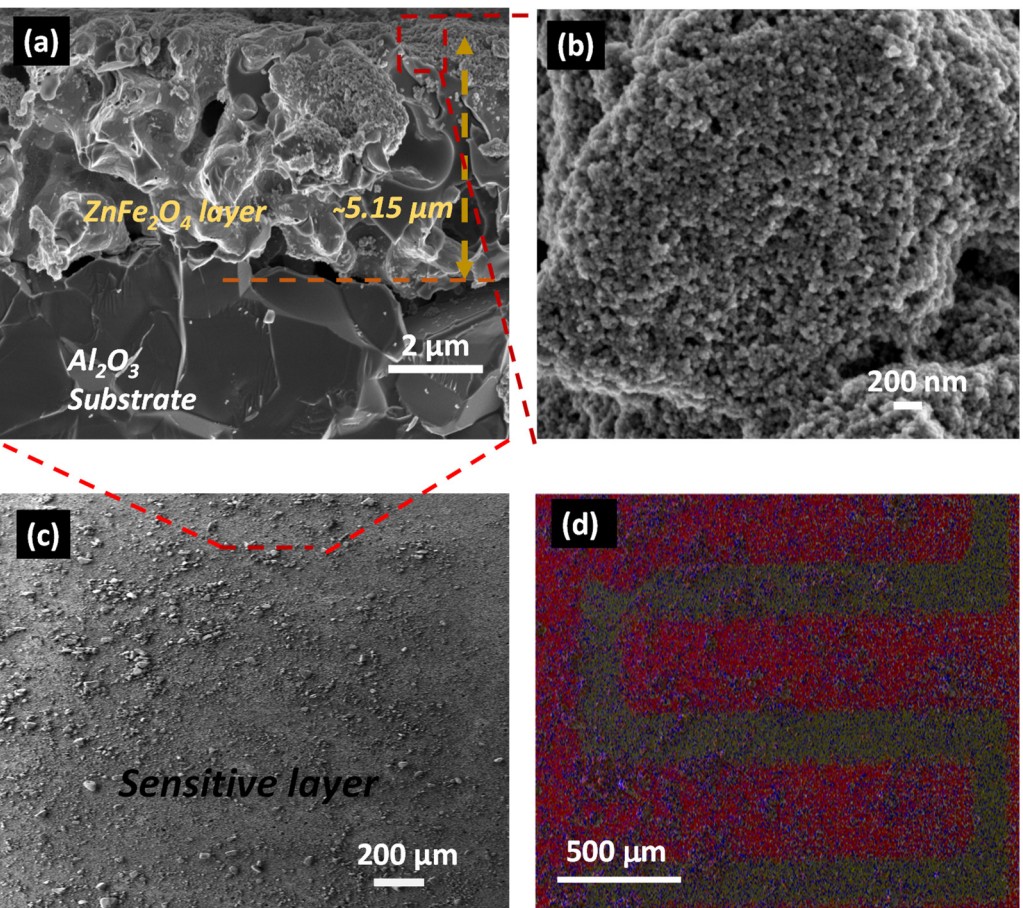

**Figure 4.** (**a**) Cross-sectional image of the $ZnFe_2O_4$ NP sensing film prepared at 2 μL. (**b**) High magnification top view of the sensing film and (**c**) top view displaying distribution of $ZnFe_2O_4$ NPs across $Al_2O_3$ interdigitated electrodes. (**d**) SEM elemental maps of the $ZnFe_2O_4$ NP sensing film.

### 3.2. Sensing Characteristics

The electrical properties of the active $ZnFe_2O_4$ NP-based sensing layers were evaluated in order to validate real applicability in sensing devices. Figure 6a presents the resistance in air of active $ZnFe_2O_4$ NP-based sensing layers obtained by varying the temperature from 25 to 180 °C. The resistance in air decreased with the increase in operating temperature, displaying the general semiconductor characteristics [29]. At operating temperatures higher than 90 °C, the resistance is not much affected by temperature change. This is probably because of the equilibrium between the competing thermal excitation of electrons and oxygen chemisorption [8]. The resistance of the active $ZnFe_2O_4$ NP-based sensing layers decreased with the volume amount with the sensing layer produced at 0.5 μL, showing a significant high resistance even at a high temperature of 120 °C. This can be associated with the reduced surface area covered by the deposited $ZnFe_2O_4$ NPs between interdigitated electrodes, resulting in reduced percolation path for charge carriers.

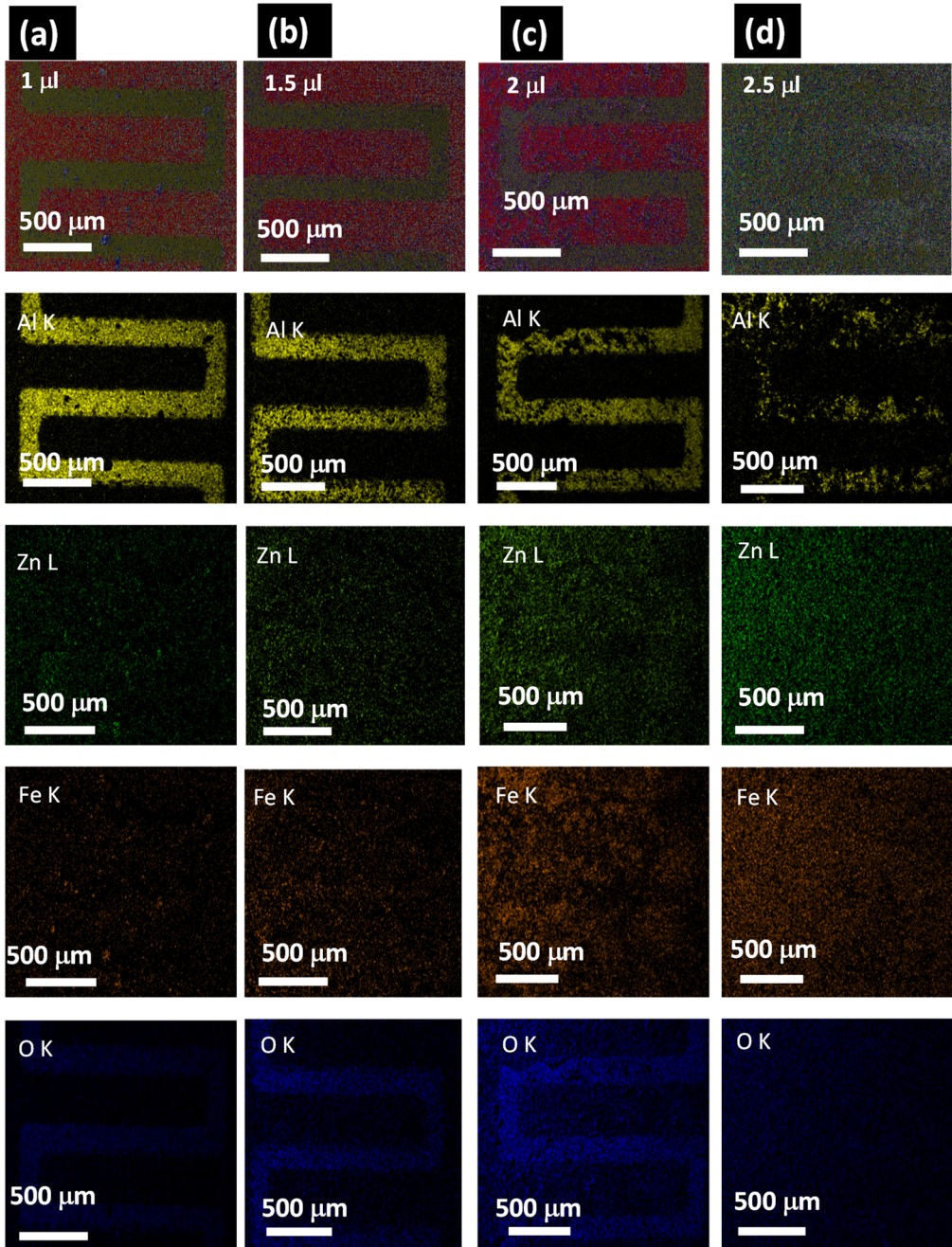

**Figure 5.** (**a**–**d**) EDS elemental mapping (Al, Zn, Fe and O) on the top view of ZnFe$_2$O$_4$ NP sensing films deposited at volumes of 1.0, 1.5, 2.0, and 2.5 μL, respectively.

The optimum operating temperature was determined by exposing the ZnFe$_2$O$_4$ NPs based sensing layers produced at volumes of 1.0, 1.5, 2.0, and 2.5 μL towards 40 ppm of propanol at operating temperatures ranging from 25 to 180 °C as shown in Figure 6b. The response values for all active ZnFe$_2$O$_4$ NP-based sensing layers increased with increasing operating temperature exhibiting an optimal response at 120 °C. This can be explained by an increase in chemical reaction and gas diffusion that varies with temperature. Beyond 120 °C, the increased excitation of electrons reduced further chemisorption of oxygen, hence the decrease in active ZnFe$_2$O$_4$ NP-based sensing layers response values. The optimum temperature of 120 °C was then chosen for conducting sensing analysis in this study.

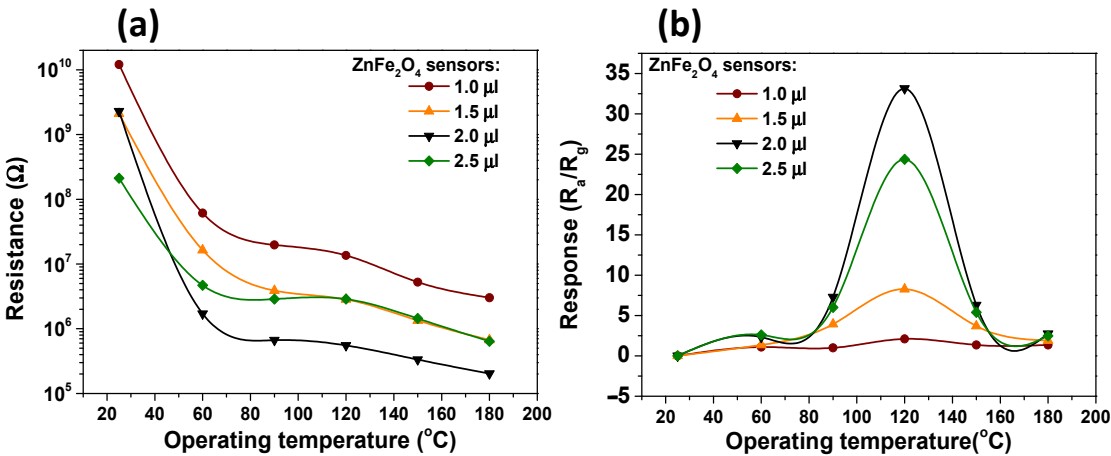

**Figure 6.** (**a**) Resistance in air and (**b**) responses of the active $ZnFe_2O_4$ NP sensing layers produced at volumes of 1.0, 1.5, 2.0, and 2.5 μL against the operating temperature ranging from 25 to 180 °C.

Figure 7a presents the transient response curves of the active $ZnFe_2O_4$ NP sensing layers produced at volumes of 1.0, 1.5, 2.0, and 2.5 μL towards 2.5 to 40 ppm propanol at an operating temperature of 120 °C. When the active $ZnFe_2O_4$ NP sensing layers produced at different volumes were exposed to various concentrations of propanol, the sensing layer responses increased rapidly and reached saturation before dropping to the base resistance. However, the active $ZnFe_2O_4$ NP sensing layer prepared at 2.0 μL displayed a higher magnitude of the response as compared to its counterparts. Figure 7b presents the calibration curves of the active $ZnFe_2O_4$ NP sensing layers towards propanol at an operating temperature of 120 °C. The response against concentration data was fitted using a power-law relation given by [30]:

$$R = R_a / R_g = (1 + k[C_3H_8O])^m \qquad (1)$$

where $k$ is the sensitivity coefficient, $[C_3H_8O]$ the concentration of propanol and $m$ the power-law exponent. The slope of the linear curve gave a sensitivity of $1.06 \pm 0.04$ ppm$^{-1}$ for the active $ZnFe_2O_4$ NP sensing layer film prepared at 2.0 μL and an extrapolated detection limit of about 0.588 ppm.

It is also well known that work efficiency is an essential parameter in gas sensing application [31]. Thus, the response and recovery times of the active $ZnFe_2O_4$ NP sensing layers produced at different volumes were measured as shown in Figure 7c. The response time was taken at a point where the sensors reach 90% of the maximum response while the recovery time was taken at 90% before baseline. Figure 7d presents response and recovery times of the active $ZnFe_2O_4$ NP sensing layer films produced at different volumes of 1.0, 1.5, 2.0, and 2.5 μL towards 40 ppm propanol. The active $ZnFe_2O_4$ NP sensing layer produced at 2.0 μL displayed better sensing kinetics than its counterparts since its response and recovery times were found to be within a minute. In addition, the active $ZnFe_2O_4$ NP sensing layer at 2.0 μL revealed a faster response time than recovery time. This could probably be due to the fact that desorption of propanol gas tends to be slower than adsorption [5,32,33].

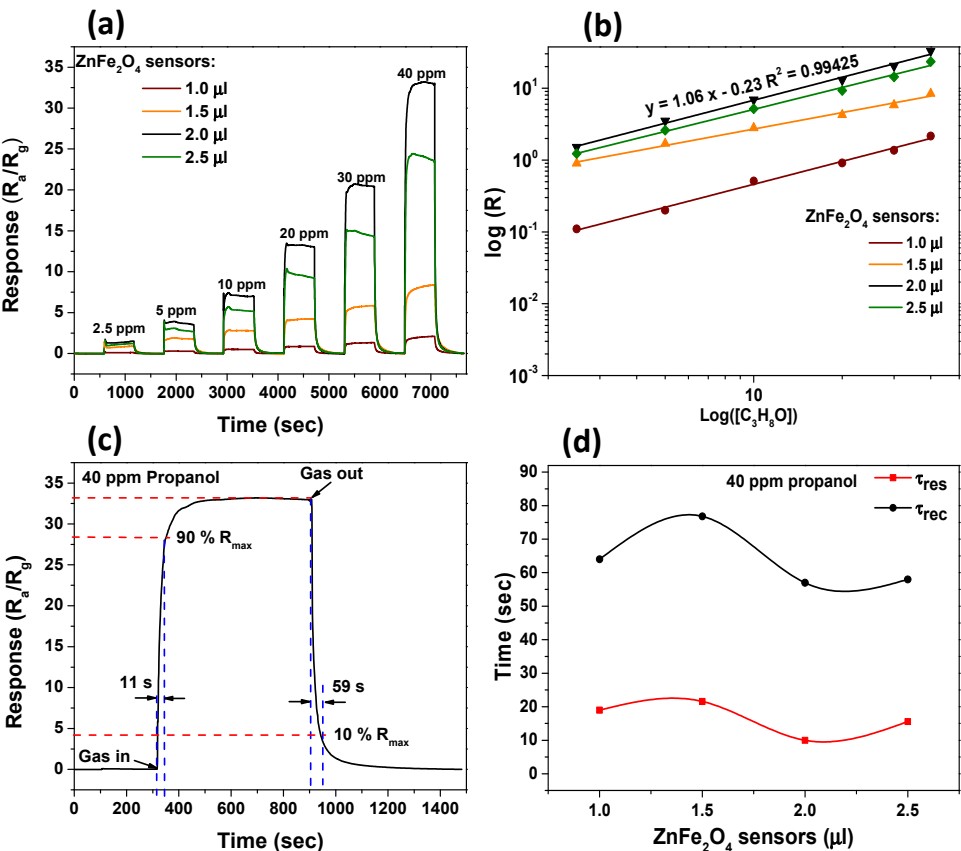

**Figure 7.** (**a**) Transient response curves, (**b**) log response against log concentration, (**c**,**d**) measurements and response-recovery times plot of the active ZnFe$_2$O$_4$ NP sensing layers produced at volumes of 1.0, 1.5, 2.0, 2.5 μL towards 40 ppm at an operating temperature of 120 °C.

In alcoholic beverages, several VOCs contribute to the sensory attributes of the drink [34,35]. Therefore, a gas sensor needs to have good selectivity when utilized for classification and quality checks in the food sector. Figure 8a compares the responses of the active ZnFe$_2$O$_4$ NP sensing layer produced at 2.0 μL towards 40 ppm of propanol, ethanol, methanol, carbon dioxide, carbon monoxide, and methane at an operating temperature of 120 °C. The comparison of responses for the active ZnFe$_2$O$_4$ NP sensing layer indicates a good selectivity towards propanol, which displayed a response value of 33 compared to other gases.

For further gas sensor practicality, the active ZnFe$_2$O$_4$ NP sensing layer's reproducibility was evaluated by repeatedly exposing the active ZnFe$_2$O$_4$ NP sensing layer produced at 2.0 μL to 40 ppm propanol at an operating temperature of 120 °C as it displayed enhanced sensing performance. Good repeatability towards 40 ppm of propanol with negligible variation in the response values was observed as presented in Figure 8b. Furthermore, an ideal sensor needs to maintain long term stability. Thus, the durability of the active ZnFe$_2$O$_4$ NP sensing layer produced at 2 μL towards 40 ppm propanol at an operating temperature of 120 °C was tested over a period of 30 days. As presented in Figure 8c, the sensor reproduced the response with limited fluctuations as shown in the inserted figure. Thus, the sensor can be relied on for long term detection of propanol.

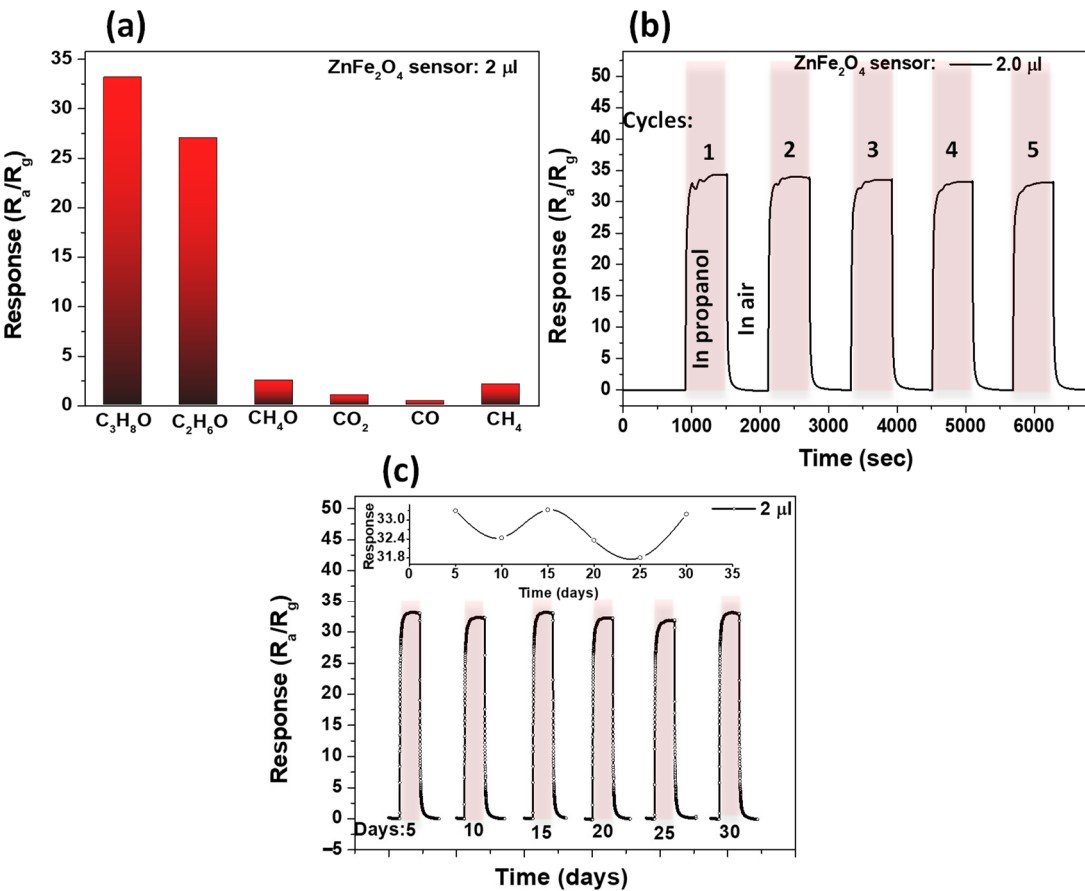

**Figure 8.** (**a**) Response comparison of the active $ZnFe_2O_4$ NP sensing layers produced at 2.0 μL to 40 ppm propanol, ethanol, methanol, carbon dioxide, carbon monoxide, and methane at an operating temperature of 120 °C in dry air. (**b**) Reproducibility and (**c**) durability curves of the active $ZnFe_2O_4$ NP sensing layer produced at 2.0 μL towards 40 ppm propanol at an operating temperature of 120 °C in dry air.

Humidity is an unavoidable parameter in the food sector in terms of food transportation and its storage. In the case of beverage alcoholics, humidity plays a role in the aging process of wine in cellars [36]. Temperature and humidity also influence the ingredients and subjective taste of wine [37]. Therefore, the active $ZnFe_2O_4$ NP sensing layers produced at different deposition volumes were exposed to 40 ppm of propanol both in dry air and atmospheric conditions of 30, 60, and 90% relative humidity (RH) levels acquired at an operating temperature of 120 °C, and the results are presented in Figure 9. As observed from Figure 9a, the resistance dropped upon an increase in RH levels. A further slight drop in resistance was observed as the RH levels were increased. This could be due to the adsorption of water molecules that displaces oxygen and forms a monolayer of –OH groups. The reduction of chemisorbed oxygen subsequently leads to the reduction of resistance. Figure 9b displays the influence of RH on the responses of the active sensing layers. As observed, the response values decreased with an increase in RH levels, and this is more evident on the active sensing layer produced at 2.0 μL. This could be explained by the competition of adsorption sites between water and oxygen species. Thus, when the sensors are exposed to propanol, there is a limited amount of oxygen species to react, leading to a drastic drop in response.

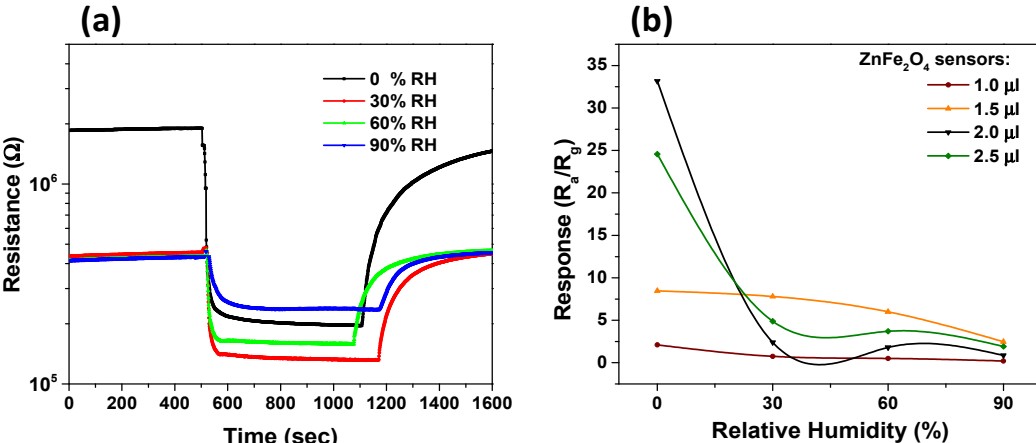

**Figure 9.** (**a**) Transient resistance curves of the active ZnFe$_2$O$_4$ NP sensing layer produced at 2.0 μL towards 40 ppm propanol at an operating temperature of 120 °C. (**b**) Response against RH of the active ZnFe$_2$O$_4$ NP sensing layer produced at different deposition volumes of 1.0, 1.5, 2.0, 2.5 μL.

## 4. Discussion

Based on the findings presented above, it is clear that the active ZnFe$_2$O$_4$ NP sensing layer produced at 2.0 μL possesses prominent sensing characteristics. The gas sensing mechanism as illustrated in Figure 10 can therefore be explained as follows: when the active sensing layer is exposed to air at an temperature of 120 °C, the adsorbed molecular oxygen dissociates to the atomic form O$^-$. The adsorbed oxygen species then retract electrons from the conduction band at the surface of the active sensing layer thus creating the so called electron depletion layer [38,39]. The thickness of the electron depleted region is the length of the band bending region. This causes an increase in the resistance of the active sensing layer. The reaction of the adsorbed oxygen species with a reducing gas consumes oxygen species at the surface layer. Depending on the concentration of the reducing gas, this allows desorption of oxygen and a release of electrons into the conduction band, resulting in decreased resistivity. For a polycrystalline structured sensing material, the single grains are in contact with their ohmic region forming a double Schottky barrier for electrons, as depicted in Figure 10. Assuming a thermal emission as the dominant transport mechanism, the conductivity over the potential barrier depends on the energy as follows [38,40]:

$$G \sim e^{-qV_s/kT} \tag{2}$$

where $qVs$ is the surface band bending, $k$ is the Boltzmann constant and $T$ is the temperature of SMO sensing layer. The active ZnFe$_2$O$_4$ NPs sensing layer produced at 2.0 μL volume has prominent sensing layer characteristics due to a high inter-agglomerate and inter-grain porous layer as seen in Figure 3 SEM images. A highly porous sensing layer which promotes both inter-agglomerate and inter-grain gas diffusion possesses a high gas-permeability and allows with a large number of grains to participate in grain-grain conductivity. This allows for more pronounced surface bending $qVs$ and leads to a high modulation of resistance and high response. Thus, the active surface is much higher than that of its counterparts due to gas access into the entire sensing layer. The enhanced sensing performance could also be due to the surface texture of the sensing layer produced at 2.0 μL. From SEM images, the sensing films produced by 1.5, 2.0, and 2.5 μL show a rough surface which may have particles oriented in preferential directions for active sites. The rough surface is also confirmed by AFM images in Figure 5. Therefore, the combination of a rough surface and gas diffusion within the active sensitive layer may be the reason of the enhanced sensing characteristics displayed by sensing layer produced at 2.0 μL.

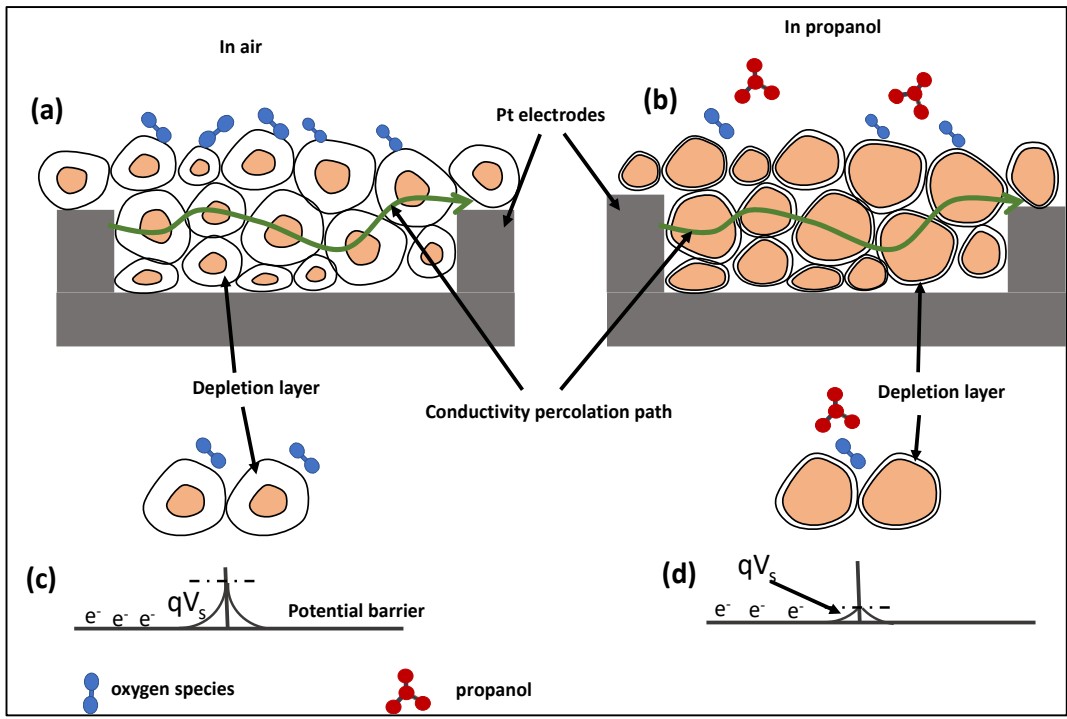

**Figure 10.** (**a**,**b**) Gas sensing mechanism of ZnFe$_2$O$_4$ NP sensing layer showing depletion layer in the presence of oxygen without propanol and in the presence of oxygen with propanol, respectively. (**b**–**d**) Charge transport of the sensing layer in the presence of oxygen without propanol and in oxygen with propanol, respectively.

A literature survey of different sensors reporting detection of propanol was conducted to compare sensing characteristics of the optimized active ZnFe$_2$O$_4$ NP sensing layer produced at 2.0 µL. The sensing characteristics of different sensors from literature are presented in Table 1. In comparison with other sensors, the active ZnFe$_2$O$_4$ NP sensing layer produced at 2.0 µL displayed a high response at relatively low concentrations towards propanol, showing potential of detecting low concentrations of VOCs in alcoholic beverages for quality checks in the food sector. Moreover, with a porous sensing layer with a high diffusion rate, the active ZnFe$_2$O$_4$ NP sensing layer produced at 2.0 µL maintained low response/recovery times as compared to other sensors deposited using similar drop-casting techniques. Thus, the active ZnFe$_2$O$_4$ NP sensing layer produced at 2.0 µL displays superior characteristics compared to other sensors from the literature.

**Table 1.** Comparison of the ZnFe$_2$O$_4$ NP sensing layer produced at 2.0 µL to other propanol gas sensing layers.

| Sensor Films | Temperature (°C) | Concentration (ppm) | Response ($R_a/R_g$) | $\tau_{res}/\tau_{rec}$ (s) | Ref. |
|---|---|---|---|---|---|
| ZnFe$_2$O$_4$/ZnO particles | 25 | 3000 | 5.2 * | 45/90 | [41] |
| ZnSnO$_3$ nanospheres | 200 | 500 | 64 | <10/315 | [42] |
| Fe$_2$O$_4$/MnO$_2$ mixtures | 25 | 5000 | 62.57 * | 60/80 | [43] |
| ZnO nanoplatelets | 125 | 40 | 6.6 | 190/200 | [44] |
| NiFe$_2$O$_4$ hollow byramids | 120 | 200 | 32.19 | - | [45] |
| ZnO@SiO$_2$/rGO spherical NP | 29 | 300 | 156.85 | 37/207 | [46] |
| ZnFe$_2$O$_4$ nanostructures | 300 | 500 | 16 | - | [47] |
| ZnFe$_2$O$_4$ NP | 120 | 40 | 33 | 11/59 | This work |

Note: Hyphen '-' values not reported. Asterisk '*' $\Delta R = (R_g - R_a/R_a) \times 100$.

## 5. Conclusions

Herein, a microwave-assisted hydrothermal method was used to synthesize $ZnFe_2O_4$ NPs. A drop-casting method was adopted and optimized to produce sensing layers at volumes of 1.0, 1.5, 2.0, and 2.5 µL. Successful deposition was confirmed by XRD patterns measured directly on the sensing layers. SEM revealed deposition of inter-agglomerate and inter-grain porous $ZnFe_2O_4$ NP sensing layers on the surface of the $Al_2O_3$ substrate. Sensing results showed that the sensing layer produced at a volume of 2.0 µL performed better towards propanol at an operating temperature of 120 °C, attaining a maximum response of 33. This was attributed to the combination of a rough surface and gas diffusion within the active sensitive layer.

**Author Contributions:** Conceptualization, methodology, funding acquisition, G.H.M. Formal analysis, investigation, data curation, writing—original draft preparation, M.I.N.; Editing, visualization, co-supervision, H.C.S. All authors have read and agreed to the published version of the manuscript.

**Funding:** This research was financially supported by the Department of Science and Innovation, South Africa (project no. C6FNS72) and the Council for Scientific and Industrial Research (Project No. HCARD03 and C1FNS98).

**Acknowledgments:** Authors would like to acknowledge Sharon Eggers for assisting with scanning electron microscope.

**Conflicts of Interest:** The authors declare no conflict of interest.

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
