# Peer review of "Enhanced Propanol Response Behavior of ZnFe2O4 NP-Based Active Sensing Layer Induced by Film Thickness Optimization"

_processes, doi:10.3390/pr9101791_

Round 1

Reviewer 1 Report

The manuscript entitled "Enhanced propanol response behavior of ZnFe2O4 NPs based active sensing layer induced by film thickness optimization", by Nemufulwi et al. presented a gas sensor using the drop-casting technique. The paper is interesting. However, there are some problems which should be addressed:

  1. In fact, these materials and techniques of the sensors are not new, although authors tried to control the thickness of the sensing layer.
  2. The sensitivity of the sensor is not high. Besides, some current studies show a low operating temperature of 75-80 °C, why this sensor has an operating temperature of 120 °C?
  3. Some other characteristics of durability, reproducibility of the sensor should be tested.
  4. A comparison with other studies (sensitivity, temperature, response/recovery time, etc.) is needed.
  5. Some minor issues:

- Why Equation 2 is bold?

- The sizes of (a), (b),... in figures are different, please fix.

Totally, authors should highlight the contributions of the manuscript.

Reviewer 2 Report

The manuscript entitled: “Enhanced propanol response behavior of ZnFe2O4 NPs based 1 active sensing layer induced by film thickness optimization” written by Murendeni I. Nemufulwi and the coautors showed how to prepare zinc ferrite (ZnFe2O4) nanoparticles in order to enhance sensing response. The materials where deposit in different thicknesses on interdigitated alumina substrates and fabricated at volumes of 1.0, 14 1.5, 2.0, and 2.5 μl using drop casting technique.  The manuscript presents a good response of the active ZnFe2O4 NPs sensing layer produced for different deposition. 

After revising the manuscript I noticed that the following changes should be considered: 

  • please clarify on the row 103-104  the appropriate amounts and what mole ratio of 1:2 ?
  • row 111 what means overnight ?
  • row 112 pistol or pestle?
  • row 117 another appropriate amounts ?
  • did you prepare yourself al2o3 substrate ? if you buy it you should mention it at materials.  Also if you did the substrate how did you manage to have the same layer thickness?
  • when you annealed at 300 oC (row 121) did all terpineol/cellulose solution burn?
  • at row 179 and 192,  1,5 μl was selected for this investigation but the for gas senzor you choose 2 μl. why? please complete the data
  • the references are not in MDPI format please correct them

Round 2

Reviewer 1 Report

Thanks for the answers. I recommend publishing this paper.

Reviewer 2 Report

Dear authors,

in the introduction part you could improve the novelty which have you made with one two phrases in order to be more accurate.